# Establishing a Novel Diagnostic Framework Using Handheld Point-of-Care Focused-Echocardiography (HoPE) for Acute Left-Sided Cardiac Valve Emergencies: A Bayesian Approach for Emergency Physicians in Resource-Limited Settings

**DOI:** 10.3390/diagnostics13152581

**Published:** 2023-08-03

**Authors:** Kamlin Ekambaram, Karim Hassan

**Affiliations:** 1Port Shepstone Regional Hospital, University of KwaZulu-Natal, Durban 4041, South Africa; 2Life Bay View Private Hospital, Mossel Bay 6506, South Africa; hsskar@gmail.com

**Keywords:** point-of-care echocardiography, handheld point-of-care ultrasound, acute valvular emergencies, acute severe aortic regurgitation, acute severe mitral regurgitation, resource-limited settings, emergency department, Bayes theorem

## Abstract

Acute severe cardiac valve emergencies, such as acute severe mitral regurgitation (AMR) and acute severe aortic regurgitation (AAR), present significant challenges in terms of diagnosis and management. Handheld point-of-care ultrasound devices have emerged as potentially pivotal tools in ensuring the prompt and accurate diagnosis of these left-sided valve emergencies by emergency physicians, particularly in resource-limited settings. Despite the increased utilisation of point-of-care ultrasound by emergency physicians for the management of patients in states of acute cardiorespiratory failure, current diagnostic protocols cannot perform sufficient quantitative assessments of the left-sided cardiac valves. This review elucidates and evaluates the diagnostic utility of handheld point-of-care focused-echocardiography (HoPE) in native AMR and AAR by reviewing the relevant literature and the use of clinical case examples from the Emergency Department at Port Shepstone Regional Hospital (PSRH-ED)—a rural, resource-limited hospital located in KwaZulu-Natal, South Africa. Combining the findings of the review and clinical case illustrations, this review proceeds to synthesise a novel, Bayesian-inspired, iterative diagnostic framework that integrates HoPE into the evaluation of patients with acute cardiorespiratory failure and suspected severe left-sided valve lesions.

## 1. Introduction

Left-sided cardiac valves play critical roles in the maintenance of cardiac output and systemic perfusion. Patients with acute severe dysfunction of these valves are more commonly present in extremis—haemodynamic instability and cardiorespiratory failure—compared to those with severe right-sided valvular lesions [1,2,3,4,5,6].

Acute left-sided native cardiac valvular dysfunction, specifically acute severe mitral regurgitation (AMR) and acute severe aortic regurgitation (AAR), represents a myriad of health concerns that contribute substantially to morbidity and mortality [2,4,7,8,9,10,11,12]. Subsequently, their clinical significance necessitates swift diagnosis and the adoption of comprehensive management strategies.

However, one of the challenges associated with the diagnosis of these conditions lies in the poor diagnostic yield of both patient history and physical examination findings of acute valve lesions, indicating the need for technological aids to diagnosis [13,14,15]. Point-of-care ultrasound (POCUS) is a bedside imaging modality that has gained popularity among emergency physicians in the evaluation of dyspnea and undifferentiated shock. However, none of the several available diagnostic protocols incorporating POCUS offer a sufficient quantification of left-sided cardiac valve function [16,17].

The use of handheld ultrasound devices (HUDs) has emerged as a more accessible, cost-effective means of complimenting cardiac examination at the point of patient care [18,19,20,21]. Accordingly, we use the term handheld point-of-care focused-echocardiography (HoPE) to refer to the specific application of goal-directed limited cardiac ultrasound at the bedside by non-cardiologists using handheld ultraportable ultrasound devices.

Subsequently, the objective of this narrative review is to create a diagnostic framework, incorporating HoPE to assist with the prompt recognition and diagnosis of AMR and AAR, in order to ensure that definitive life-saving therapy can be instituted timeously. To accomplish this, we describe and detail left-sided cardiac haemodynamics and elucidate and evaluate the diagnostic utility of HoPE for AMR and AAR in acute cardiorespiratory failure. We perform these tasks by reviewing relevant literature and conducting retrospective analysis of clinical cases from the emergency department (ED) at Port Shepstone Regional Hospital (PSRH-ED)—a rural, resource-limited hospital located in KwaZulu-Natal, South Africa. We then synthesise, present, and explain the proposed HoPE diagnostic framework.

Notably, while prosthetic valve dysfunction might also present with an equal degree of severity as native valve dysfunction, the bedside assessment of prosthetic heart valves with transthoracic echocardiography can be difficult—often requiring a more in-depth assessment using transoesophageal echocardiography (TOE)—and is beyond the scope of this review [4,22].

## 2. Methods

### 2.1. Search Strategy

We performed a comprehensive search of the PubMed, Embase, and Cochrane databases for English language articles published between 1980 and 2023. We carried out this task using a combination of medical subject heading (MeSH) terms and keywords, including “cardiac haemodynamics”, “left side”, “echocardiography”, “point-of-care”, “handheld”, “portable”, “education”, “acute cardiac valve emergencies”, “diagnostic framework”, “diagnostic protocol”, “Bayes’ theorem”, “iterative diagnostics” and “emergency department”. The search was also supplemented by conducting a manual reviewing of the references of identified articles in order to locate additional relevant studies.

### 2.2. Synthesis of Results

A narrative synthesis approach was used to summarise the findings from the included papers by topics of interest.

### 2.3. Clinical Cases

To contextualise our findings, we include two illustrative clinical cases from our institution that demonstrate the utility of HoPE in diagnosing and managing acute cardiac valvular emergencies. These cases are presented in accordance with the CARE guidelines [23] for clinical case reporting, and patients’ informed consent was obtained for this.

### 2.4. Synthesis of a Diagnostic Framework

Based on the findings of the review and clinical case illustrations, we synthesised a Bayesian-inspired, iterative diagnostic framework that incorporated HoPE for the rapid assessment and disposition planning of patients with suspected acute left-sided valvular dysfunction in the ED. The proposed algorithm was reviewed by specialists in emergency medicine, cardiology, and process engineering for face validity.

## 3. Results

### 3.1. Left-Sided Cardiac Haemodynamics

The left-sided cardiac valves contribute significantly to maintaining haemodynamic homeostasis in the human circulatory system by ensuring the unidirectional flow of oxygen-rich blood from the lungs, through the left-sided cardiac chambers, and finally to the systemic circulation [15,24,25]. The modulation of this flow is achieved through a meticulously orchestrated process where, at any given moment, only two chambers—the left atrium (LA) and left ventricle (LV), or the LV and aorta—are in communication. This preserves an unceasing forward propulsion of blood across pressure gradients [24,26]. The clinical sequelae of left-side valve incompetence relate directly to the ability of the heart to accommodate pathological states through remodeling (Figure 1).

The contrasting presentations of acute and chronic severe valvular incompetence illuminate the intrinsic complexity of cardiac haemodynamics and demonstrate how an acute disruption in this delicate equilibrium could swiftly result in a lethal cascade of pathophysiological events, strongly emphasising the importance of prompt recognition and institution of definitive, and potentially lifesaving, therapy (Figure 2).

### 3.2. Chronic Left-Sided Valve Incompetence

The chronic phenotype evolves insidiously over an extended period of time that often spans several years or decades [24]. The progressive regurgitant pathophysiology described above triggers compensatory cardiac mechanisms, principally left ventricular dilation and hypertrophy. These work for a time to preserve normal cardiac function. Consequently, the initial stages may be clinically silent, while subsequent manifestations include fatigue, exertional dyspnea, palpitations, and decompensated heart failure. Management is multifaceted, encompassing pharmacotherapy for symptomatic relief and heart failure risk mitigation. Surgical interventions, including valve repair or replacement, are reserved for severe symptomatic cases or asymptomatic individuals with high-risk features [2].

### 3.3. Acute Severe Left-Sided Valve Incompetence

In stark contrast, the acute phenotype is precipitated by a sudden pathophysiological event such as endocarditis, papillary muscle rupture, or aortic dissection. The abrupt onset of excess volume-loading precludes compensatory ventricular compliance, thus resulting in significant pressure changes within the left-sided cardiac chambers [24].

For example, during systole, under physiological circumstances, the LV engages exclusively with the high-pressure aorta facilitated by a closed mitral valve (MV). However, with the pathological development of AMR, the LV communicates concurrently with the high-pressure aorta and the low-pressure LA. As the volume of blood flow is proportional to the pressure gradient between two the cardiac chambers and tends to flow in the path of least resistance, the bulk of the LV stroke volume flows backwards in preference of the low-pressure LA to the high-pressure aorta, subsequently compromising forward systemic flow [7,8,24]. Similarly, with AAR, the regurgitant volume from the aorta into a non-compliant LV leads to a sudden rise in LV pressure during diastole, leading to the degradation of the diastolic pressure gradient between LA and LV. This in turn compromises diastolic LV filling and reduces cardiac output [3,8].

Thus, the mechanics of AAR and AMR result in typical presentations of profound dyspnea, cardiorespiratory failure, and shock, necessitating immediate clinical stabilisation and potentially urgent surgical correction of the issues causing regurgitation.

### 3.4. Echocardiography

A comprehensive echocardiogram is a type of cardiac ultrasound, typically performed by a cardiologist or trained sonographer, that is regarded as the gold standard for diagnosing acute severe valve lesions. Over the decades, echocardiography has seen a transformative journey from large, low-resolution machines to two-dimensional brightness (2D B) and motion (M) mode devices, Doppler echocardiography, TOE, and three-dimensional echocardiography. These changes have enhanced resolution, anatomical detail, and the ability to perform non-invasive blood-flow measurements.

Modern trends have focused on portability, giving rise to HUDs that can connect to smartphones or tablets, thereby making cardiac imaging more accessible and redefining point-of-care diagnostics [27,28,29,30,31]. Importantly, despite significant advancements in HUD technology, the majority of currently available devices lack continuous-wave Doppler (CWD), limiting their suitability for use in comprehensive echocardiographic exams.

Correspondingly, another type of cardiac ultrasound, known as focused cardiac ultrasound (FOCUS), has emerged. Typically performed by non-cardiologists, this is a limited, goal-directed method perfectly suited for use in time-critical bedside applications [32,33].

### 3.5. Handheld Point-of-Care Focused-Echocardiography (HoPE)

To elucidate terminology and to differentiate between various forms of cardiac ultrasound, it is important to understand that FOCUS is a limited form of comprehensive echocardiography—which is a type of cardiac ultrasound. FOCUS, which is increasingly performed by non-cardiologists, is distinctive due to its more qualitative, goal-directed approach that typically does not require advanced imaging modes or comprehensive knowledge of transthoracic cardiac ultrasound image interpretation [17,34,35,36].

The European Society of Cardiology (ESC) has outlined competency requirements for FOCUS to ensure that practitioners adhere to specific standards [37]. These standards include theoretical knowledge, practical skills, supervised scanning sessions, and competence assessment. These principles can be applied to HoPE for the assessment of valvular pathology—requiring first a foundational understanding of the following:A.Basic Cardiac Views (Figure 3)The standard echocardiographic views required to assess cardiac valvular structure and function include the parasternal long-axis (PLAX), parasternal short-axis (PSAX), apical four-chamber (A4C) and subcostal (SC) views. These views facilitate the visualisation of the aortic, mitral, tricuspid, and pulmonic valves, thus enabling the identification of valvular abnormalities [38].

B.Imaging ModesVarious ultrasound modalities are used to assess the structure and function of heart valves. Two-dimensional (2D) imaging is generally used for structural assessment, colour flow doppler (CFD) is used to evaluate blood flow across the valves, and continuous-wave (CWD) and pulsed-wave doppler (PWD) techniques are used to measure flow velocities and gradients across valves. Integrating these imaging modalities can offer a comprehensive assessment of valvular function and aid in identifying specific pathologies (Table 1) [27,38].

C.Clinical Integration

Emergency physicians should utilise critical thinking and diagnostic reasoning to integrate cardiac ultrasound findings (Table 1) with the patient’s history, findings of clinical examinations and laboratory results to determine the most probable underlying diagnosis and to guide further management. The pre-test or prior probability of specific valvular disorders should be considered, and HoPE should be used to increase or decrease the likelihood of the suspected diagnosis [27,39].

**Table 1 diagnostics-13-02581-t001:** Characteristic echocardiographic findings in acute mitral and aortic regurgitation [1,7,8,27,40,41].

Echocardiographic Technique	Acute Mitral Regurgitation	Acute Aortic Regurgitation
2D B-Mode		
Valve Morphology	Flail, prolapsing or perforated leaflet. Ruptured chordae tendineae. Ruptured head of papillary muscle. Vegetations.	Root dilatation and dissection flaps. Vegetations. Torn valve cusps (usually due to trauma).
Left Atrium	Variable. Could be of normal size if pure acute severe MR. Usually enlarged with any degree of chronic MR.	Variable. Normal-size if pure acute severe AR. Usually enlarged if any degree of chronicity
Left Ventricle	Normal-size if acute severe MR. Dilated if acute-on-chronic severe MR.	Normal-size if acute severe AR. Dilated if acute-on-chronic severe AR.
M-Mode		Premature diastolic closure of the mitral valve.
Colour Flow Doppler (CFD)	Colour flow is not a good marker for assessment of severity of MR or AR. However, it may assist with determining aetiology—usually eccentric jets if prolapse/flail in the direction that is away from the prolapsing/flail leaflet i.e., anterior mitral valve leaflet prolapse leads to a posteriorly directed jet.	Central jet, typically directed towards the posterior wall of the left ventricle.
Continuous Wave Doppler (CWD)	Increase in forward flow velocity with “V-wave” cut off. Equal colour density between forward flow and regurgitant doppler traces. Systolic flow reversal in 3 out of 4 pulmonary veins.	Increase in forward flow velocity. Pressure half time <200 ms. Doppler trace of AR ends before the end of diastolic due to early equilisation of aortic and LV pressures. Equal colour density between forward flow and regurgitant doppler traces.
Pulse Wave Doppler (PWD)	Systolic reversal of flow in pulmonary veins.	Diastolic flow reversal in the descending thoracic aorta.

### 3.6. POCUS Protocols

The use of standardised diagnostic pathways and protocols that incorporate POCUS can significantly enhance the effectiveness and reliability of acute patient care [42]. These protocols not only provide a clear and structured learning pathway for clinicians, but also foster efficient documentation and communication of findings among health care provider, thereby elevating the overall quality of patient care and safety [39].

The adoption of a uniform approach to performing cardiac ultrasound examinations among different emergency physicians has the potential to reduce variability in image acquisition and interpretation, thus strengthening the validity of results [39]. Furthermore, the protocol-oriented approach likely quickens the diagnostic process, a vital asset in treating time-sensitive situations such as acute severe left-sided valve lesions.

A review of the trends in POCUS protocols for the emergency and critical care settings by Fan et al. [20] in 2022 found the following standardised protocols for patients presenting with acute dyspnea and/or undifferentiated shock: the bedside lung ultrasound for emergency (BLUE) protocol; the focus-assessed transthoracic echocardiography (FATE) protocol; the critical care chest ultrasonic examination (CCUE) protocol; and the rapid ultrasound in shock (RUSH) protocol. While most of these protocols include a qualitative assessment of global cardiac function, none include an assessment of cardiac valve function.

### 3.7. HoPE for Left-Sided Cardiac Valve Evaluation

In 2010, a consensus statement of the American Society of Echocardiography (ASE) and the American College of Emergency Physicians (ACEP) noted, during the evaluation of the patient with dyspnea, that “… the presence of significantly stenotic valves or regurgitant lesions using 2D and color Doppler techniques may be suggested by a FOCUS [but] full evaluation requires the quantitative analysis provided by a comprehensive echocardiogram [43].”

Since then, there has been a dearth of published literature on the intrinsic diagnostic accuracy of HUD for AMR and AAR. Nevertheless, Arntfield et al. [41] and Millington et al. [44] concluded that PoCE served as an indispensable instrument for the diagnosis and treatment of acute cardiac conditions, including valvular emergencies. Similarly, Johnson et al. [45] affirmed the value of PoCE as a vital apparatus for emergency physicians assessing patients in shock.

Further supporting this, Mukherjee et al. [46] emphasised the necessity of integrating echocardiography into the diagnostic repertoire of emergency physicians when reporting a case of papillary muscle rupture masquerading as sepsis. This was reinforced by both Bustam et al. [47] and Frederiksen et al. [48], who reported that after brief training, emergency physicians were able to accurately perform and interpret PoCE, substantially altering the course of patient care.

When researching the utility of treatments, we found a review by Chamsi-Pasha et al. [34] which suggested that the diagnostic accuracy of HUD for valvular heart disease is roughly 80%. However, a contrasting study by Marbach et al. [49] reported that the FOCUS-assisted clinical exam was poorly sensitive for aortic (46%) and mitral (71%) valve disease.

Numerous studies in low-resource settings [17,18,19] report a high sensitivity for confirming definitive rheumatic heart disease (91–97%) when groups of pediatric cardiologists, cardiology fellows and echocardiography technicians use both cart-based point-of-care echocardiography (PoCE) and handheld point-of-care echocardiography (HH-PoCE) in children. Specifically, Beaton et al. [17] reported that pediatric cardiologists using HH-PoCE detected pathological mitral regurgitation with a sensitivity of 83.3% and specificity of 100%. However, due to a lack of CWD, the subjective grading of severity of mitral regurgitation was noted to only be 77% sensitive and 69% specific.

Sonography in hypotension and cardiac arrest (SHoC) investigators performed a prevalence study during an interim analysis of two ongoing studies. Their work reported that an “Abnormal Valve Function” accounted for 39% of patients (*n* = 151) presenting to the emergency department in cardiac arrest, while valvular pathology did not feature as a cause of hypotension [50]. Subsequently, in one of the largest randomized controlled trials evaluating the utility of POCUS performed by trained practitioners on patients (*n* = 273) presenting with undifferentiated shock to emergency departments in North America and South Africa [21], the same investigative group found cardiac-related final diagnoses across both groups to be:Left ventricular failure (8%),Dysrhythmia (2%),Cardiac tamponade (1%).They reported no instances of acute valvular dysfunction.

### 3.8. Medical Diagnostics

Given the skill level and device capability of non-cardiologists with HoPE, the intrinsic diagnostic accuracy for detecting AMR and AAR is likely to be low and, therefore, an exploration of medical diagnostics is paramount before designing a diagnostic framework.

Bayes’ theorem [51] is a fundamental principle in statistical inference that provides a mathematical framework for updating beliefs about a hypothesis based on the available evidence—as occurs in medical diagnostics [52]. The theorem plays a critical role in understanding and refining the diagnostic process, especially for rare and potentially fatal conditions such as acute severe left-sided valve emergencies. In this context, pre-test probability usually represents the underlying prevalence of these emergencies in the patient presenting with acute cardiorespiratory failure, while post-test probability represents the likelihood of such an emergency given a positive result on diagnostic tests such as echocardiography.

Clinical decision rules are evidence-based diagnostic frameworks that improve diagnostic accuracy by integrating individual patient or test characteristics with clinical indicators to enhance the probability assessment of disease, thus helping to streamline the diagnostic process. Similarly, iterative diagnostics is the process of simultaneously forming and testing hypotheses, often simultaneously applying intuitive and analytic reasoning where feedback of posterior (post-test) probability becomes the new prior (pre-test) probability. Fundamental to this process is the understanding of test and treatment thresholds [53,54].

Test and treatment thresholds are two key probability estimates in medical decision-making based on Bayesian logic. The test threshold is the pre-test probability below which a disease, such as AMR or AAR, is deemed so unlikely that additional diagnostic testing is not considered cost-effective or may potentially be harmful. An example of this is a patient presenting with dyspnea, polyuria, and polydipsia with a history of diabetes who ran out of insulin. While the probability of acute left-sided valve disease here is not zero, given the clinical information, a simple bedside clinical examination of the cardiovascular system may be enough to rule out acute severe left-sided valve disease as the cause of dyspnea.

Conversely, the treatment threshold is the pre-test probability above which the disease likelihood is so substantial that the risks and costs associated with further testing outweigh the benefits, thereby prompting immediate initiation of treatment. In the example above, a single blood gas analysis showing a high blood glucose level and increased anion gap metabolic acidosis is likely enough to determine the presence of diabetic ketoacidosis (DKA) and start treatment without having to measure serum beta hydroxybutyrate levels [55]. Importantly, these thresholds are dynamic and depend on several factors: disease progression, the inherent risks of testing and treatment, the patient’s overall clinical status, and the patient’s individual values and preferences.

Furthermore, interpreting these thresholds also involves understanding the post-test probabilities represented by the positive and negative predictive values (PPV and NPV) during the diagnostic process [52]. The PPV refers to the probability of a disease being present given a positive test result (how likely is the diabetic patient described above to have AMR if systolic color flow is noted to emanate from the left atrium into the left ventricle on HoPE assessment?), while the NPV refers to the probability of disease absence given a negative test result (can practitioners rule out a myocardial infarction as the precipitant of DKA using HoPE alone?).

When the pre-test probability is below the test threshold, the disease likelihood is so low that a positive result is more likely to be a false positive, yielding a low PPV. Conversely, when the pre-test probability exceeds the treatment threshold, the disease likelihood is so high that a negative result is likely to be a false negative, resulting in a low NPV. For pre-test probabilities between the test and treatment thresholds, diagnostic testing is conducted to improve the accuracy of the disease probability, consequently enhancing both the PPV and NPV. Thus, iterative testing allows the clinician to constantly update the probability of disease until one of these thresholds are breached. During acute undifferentiated critical illness, clinicians often test multiple hypothesis concurrently to narrow the list of beneficial therapeutic interventions available.

We now reflect on two cases from our ED in which it was likely that iterative diagnostics and Bayesian logic were concurrently employed in a resource-poor setting without onsite cardiology but with access to cart-based PoCE (M7 Premium, Mindray, Nanshan Shenzhen, People’s Republic of China). The specialist emergency physician performing the point-of-care studies was trained in echocardiography.

Case 1: Acute Mitral Regurgitation due to Flail Leaflet

A 65-year-old female presented to PSRH-ED with 2-day history of chest pain and shortness of breath. Her background history is significant in that she had mMRC (Modified Medical Research Council) grade 3 chronic obstructive lung disease (COPD) and had undergone an intensive care admission for intravenous thrombolysis following a massive unprovoked pulmonary embolism (PE) 5 years prior. On examination, she was haemodynamically stable with a pansystolic murmur best audible at the apex. Electrocardiogram (ECG) showed evidence compatible with a completed posterior infarction. Troponin I returned positive.

PoCE revealed an akinetic posterior wall along with an eccentric, anteriorly directed jet of mitral regurgitation (Figure 4).

She was discharged in a stable condition following a short stay in our coronary care unit where a confirmatory echocardiogram performed by a trained on-site sonographer confirmed the presence of wall motion abnormalities and reported the mitral regurgitation as being mild. The in-patient service also performed a computed tomography pulmonary angiography (CTPA), which noted interlobular septal thickening and found no evidence of pulmonary thromboembolic disease. The patient was treated for acute coronary syndrome with concomitant pneumonia and declined transfer to the dedicated cardiac unit at our quaternary referral hospital for invasive cardiac imaging.

One month later, she re-presented to PSRH-ED in cardiorespiratory collapse. She was confused and peripherally cyanosed with a heart rate of 152 bpm, respiratory rate of 40 bpm, and blood pressure of 78/54 mmHg. Given the history of her prior admission and PoCE findings, despite her background of COPD, PE, coronary artery disease and diagnosis of pneumonia, there remained a strong clinical suspicion (high pre-test probability) of AMR.

This time, PoCE revealed a flail posterior mitral valve (PMVL) leaflet in 2D in both the A4C (Figure 5a,b) and PLAX (Figure 5c,d) views. In addition, a mobile mass adherent to the tip of the PMVL flicked in and out of the LA, which appeared to be a ruptured head of a papillary muscle on further interrogation (Figure 5b). These findings were compatible with AMR secondary to papillary muscle rupture. She was aggressively resuscitated and expeditiously transferred to the nearest cardiothoracic surgery unit.

Case 2: Acute Severe Aortic Regurgitation due to Infective Endocarditis

A 45-year-old male with end-stage renal disease, who had undergone insertion of a permanent haemodialysis in hospital two weeks prior, presented to PSRH-ED with sudden-onset dyspnea and chest pain. On examination, he was tachycardic (HR 110 bpm), pyrexial, and in respiratory distress (RR 28 bpm). There were also clinical findings compatible with cardiorespiratory failure. He had a bounding brachial pulse, but no audible murmur.

Given the clinical features of both systemic infection and acute cardiorespiratory failure, acute left-sided valvular dysfunction was suspected.

PLAX acquisition on PoCE showed a large independently mobile mass lesion adherent to the aortic valve on 2D B-mode (Figure 6a), which was highly indicative of a vegetation.

CFD on PLAX (Figure 6b) showed a central jet of regurgitant flow from the aorta into the left ventricle, which occupies the entire left ventricular outflow tract. Finally, PWD applied to the abdominal aorta (Figure 6c) showed early diastolic flow reversal, with mid-to-end diastolic cut-off. These findings were consistent with the patient symptomatology, meeting the treatment threshold, and confirmed the diagnosis of AAR secondary to infective endocarditis (IE).

A second HoPE study was performed by the same operator using the Butterfly iQ+ (Butterfly Network, Inc, Burlington, MA, USA) for comparison (Figure 6d,e) after instituting appropriate management and urgent referral for surgical intervention.

### 3.9. Synthesis of a Novel, Iterative, Diagnostic Framework for Acute Severe Left-Side Valve Emergencies

By combining the findings of the review, clinical case illustrations, and the diagnostic capabilities of HUD for HoPE, we synthesised a novel framework for use in the assessment of the patient suspected of having an acute severe left-sided valve emergency (Figure 7). The framework pragmatically integrates the clinical assessment of the patient in acute cardiorespiratory failure with HoPE into an iterative Bayesian diagnostic pathway that is streamlined for use in the average ED.

Thus, entrusting the emergency physician to consider the following three questions when applying the framework and interpreting HoPE is fundamental to our approach [56]:Is there severe valve dysfunction?Does the identified valve dysfunction explain the current clinical condition of the patient?How does this finding contribute to the patient’s management plan?

The diagnostic framework begins with the identification of a patient in cardiorespiratory failure. While initiating standard resuscitation efforts, the emergency physician simultaneously gathers and interrogates the dynamic clinical data. The physician then, based on diagnostic acumen, formulates a prior probability for left-sided valve disease, and—if this estimate breaches the test threshold—proceeds to evaluation with HoPE.

HoPE is then used to evaluate the aortic and/or mitral valves, an assessment of severity is correlated (as shown in Table 1), and the following are implemented accordingly:If the treatment threshold for either AMR or AAR is reached, the physician (or treating team) simultaneously contacts a specialty service while instating context specific haemodynamic resuscitative measure—the discussion of which are beyond the scope of this review.If the treatment threshold is not reached, the iterative process prompts the physician to re-evaluate the probability of left-side valve disease; if it continues to breach the test threshold, the physician continues to test. Alternatively, the physician has to consider other diagnoses.

Importantly, this process is not disposable and should be revisited in the face of new evidence, such as additional or supplementary patient history or a more skilled HoPE operatory, or in response to treatment or interventional outcomes.

## 4. Discussion

The precise incidence of acute severe left-sided regurgitation is difficult to ascertain as patients presenting with acute cardiorespiratory collapse are typically excluded from trails and registries. However, an examination of the prevalence or precipitating causes—rheumatic heart disease complicated by infective endocarditis and mechanical complications of myocardial infarction—suggests that the prevalence of AAR and AMR are likely higher in developing countries where the health impact of these devastating conditions are likely to be underrecognised given the lack of reliable diagnostic modality [24,57].

Encouragingly, the use of technological aids to diagnosis, such as POCUS, are gaining popularity as adjuncts to the physical exam in low-resource settings [13,14,18,24,58,59]. Yan et al. [60] reported that, after only a six-hour training program in FOCUS, medical students correctly identified 54 out of the 126 significant valve lesions identified using comprehensive echocardiography. While the quantitative assessment of valvular heart disease is limited by a lack of a comprehensive doppler package on most available HUD [61,62], Sachpekidis et al. [63] studied the CWD capability of a novel HUD (Kosmos; EchoNous Inc., Redmond, WA, USA) and concluded that it not only detects clinically significant aortic stenosis but can also be used to facilitate severity grading. Although the imminent introduction of CWD is an exciting development for HUD, it requires advanced echocardiography training to acquire and interpret data and its use is likely beyond the scope of practice for most emergency physicians in resource poor settings [43].

Consequently, we hypothesised that, by incorporating the limited doppler package of modern HUD devices into an iterative diagnostic framework, physicians may be able to improve their diagnostic accuracy of acute severe left-sided valve disease. After reflecting on the case studies presented—AMR due to a ruptured papillary muscle (Case 1) and AAR due to infective endocarditis (Case 2)—we surmised that this approach is potentially non-inferior to ED performed HoPE.

Interestingly, the SHoC-ED trial—which compared patient outcomes when radomised to POCUS plus standard care versus standard care without POCUS—found no instances of acute valvular catastrophe as a cause of hypotension despite studying over 250 patients across two continents [21]. While the exclusion of dyspneic patients without hypotension is one possible cause, it is more likely that valvular pathology was overlooked due to the limitations of POCUS in 2017 and since there were no confirmatory echocardiographic exams performed. The investigators combined features of the RUSH and ACES exams into a standardized protocol. This included the basic cardiac views but lacked the application of color doppler. Based on the findings of our review, it is possible that the findings of a hyperdynamic left ventricle with morphologically normal chambers may have led providers to assume sepsis as most likely cause in a patient with acute severe left-sided cardiac disease.

We synthesised our Bayesian-inspired, iterative diagnostic framework, leveraging HoPE by interrogating and assimilating an understanding of left-side cardiac haemodynamics with the pathogenesis and pathophysiology of acute left-side valve incompetence. The use of Bayesian logic in diagnostic medicine is well studied [51,64,65], and the combination of a Bayesian process with iteration likely amplifies the diagnostic accuracy and utility of the framework by constantly requiring and ensuring that the clinician amalgamate clinical and diagnostic information with the test and treatment thresholds in mind.

Another important finding of this review article is the identified necessity of a structured HoPE training program to ensure competency among practitioners [16,19,33,66,67]. The proposed core requirements for the clinical integration of HoPE provide a roadmap of how to elevate the skills and knowledge of physicians in dealing with acute cardiac emergencies, an approach that could be adopted by other healthcare facilities in similar settings [37].

Our review also highlights the increased accessibility and portability of HUD. These features make HoPE ideally suited for use in resource-limited settings like South Africa, where access to more advanced diagnostic imaging modalities and/or cardiology services might be scarce [68,69,70]. While challenges persist in resource-limited settings—such as limited access to equipment, inadequate training, and inconsistent image quality—ongoing efforts to improve HoPE ubiquity and enhance training programs can mitigate these issues and improve patient outcomes in acute valvular emergencies.

Penultimately, while this review offers a novel diagnostic framework for patients with acute cardiorespiratory failure suspected of acute severe left-sided regurgitant valve lesions, prospective studies are required for validation, implementation and protocolisation. Out intension is not for the framework to be used a standalone tool for the assessment of the patient in acute cardiorespiratory failure, but rather for it to compliment already-established POCUS protocols. Further research is also required to assess the framework’s performance in the diagnosis of other cardiovascular emergencies such as acute presentations of chronic severe valve disease, right-sided valve emergencies, and mechanical complications of myocardial infarction.

Finally, the cases presented above as proof of concept for the use of HoPE were performed by at a single site by a single provider trained in echocardiography. This limits the generalisability and validity of HoPE—a topic for further inquiry.

## 5. Conclusions

This review proposes a novel iterative Bayesian-inspired diagnostic framework, incorporating handheld point-of-care focused-echocardiography (HoPE) in order to improve the time required to begin the definitive treatment of acute severe left-sided cardiac valve emergencies. This constitutes a crucial step toward reducing the morbidity and mortality of these devastating conditions—particularly in low-resource settings. Despite face validity, prospective studies are required to validate this framework, study its integration into existing pathways of care, and determine its potential impact on patient care.

## Figures and Tables

**Figure 1 diagnostics-13-02581-f001:**
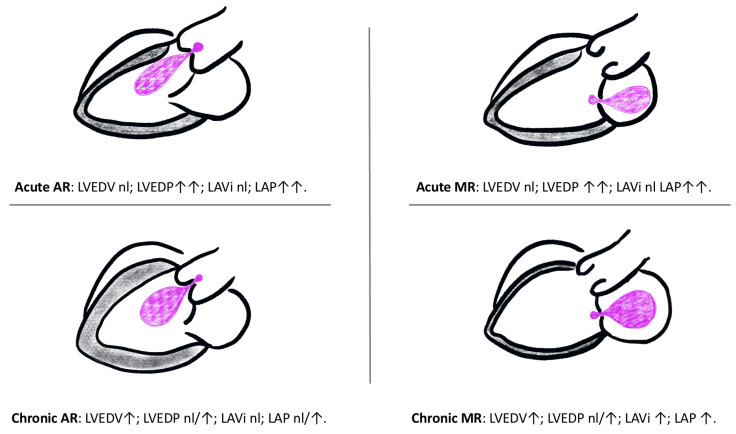
Haemodynamic consequences in acute and chronic valvular regurgitation. LAP left atrial pressure; LAVi, left atrial volume index; LVEDV, left ventricular end-diastolic volume; LVEDP, left ventricular end-diastolic pressure; nl, normal; ↑, elevated; ↑↑, significantly elevated [15].

**Figure 2 diagnostics-13-02581-f002:**
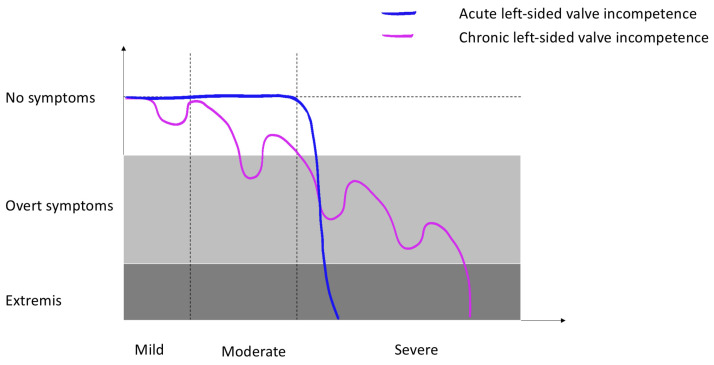
Natural history of left-sided valve lesions relating symptomatology and disease progression.

**Figure 3 diagnostics-13-02581-f003:**
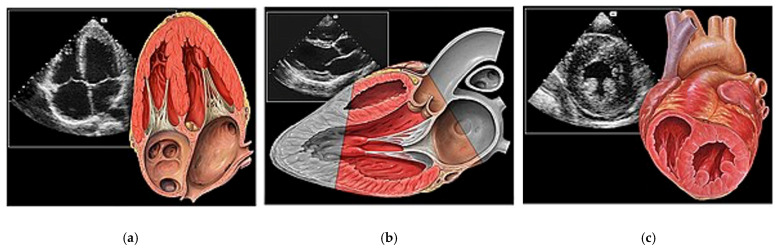
Essential cardiac windows for left sided valvular assessment using a phased array ultrasound probe: (**a**) Apical 4 Chamber (A4C), (**b**) Parasternal Long Axis (PLAX) and (**c**) Parasternal Short Axis (PSAX). Image credits: Patrick J. Lynch, medical illustrator; C. Carl Jaffe, MD, cardiologist. https://creativecommons.org/licenses/by/2.5/ (accessed on 16 May 2023).

**Figure 4 diagnostics-13-02581-f004:**
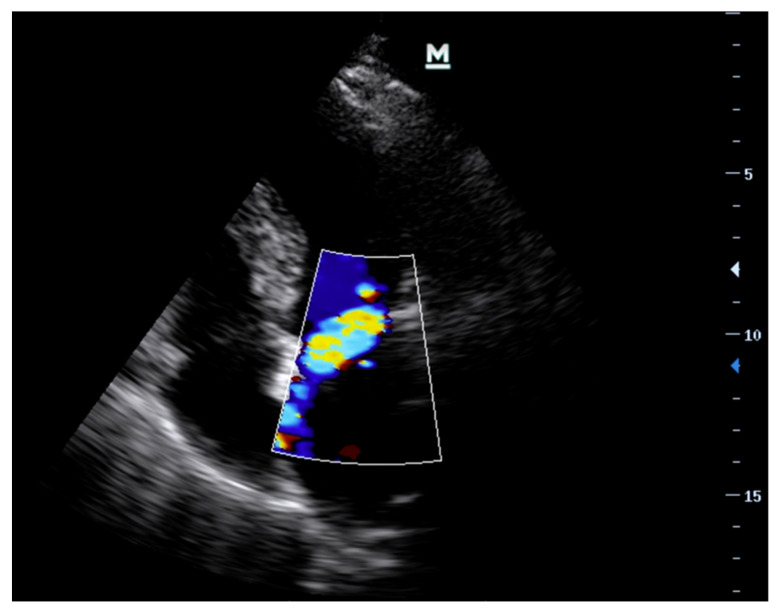
Case 1: Initial PoCE image of the A4C view with CFD.

**Figure 5 diagnostics-13-02581-f005:**
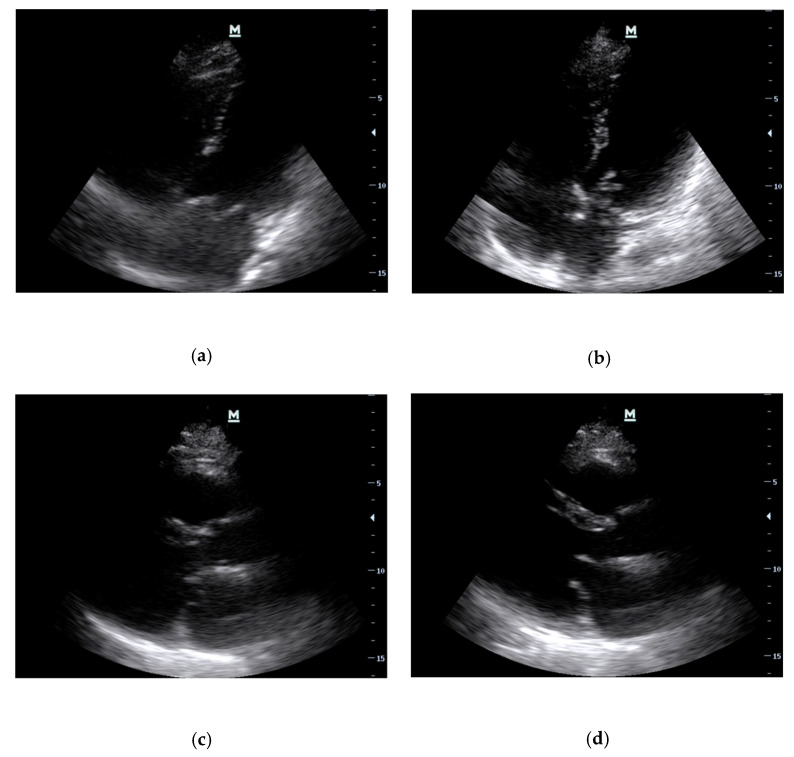
Case 2: Representation point-of-care echocardiography study showing the A4C view in systole (**a**) and diastole (**b**); PLAX view in systole (**c**) and diastole (**d**).

**Figure 6 diagnostics-13-02581-f006:**
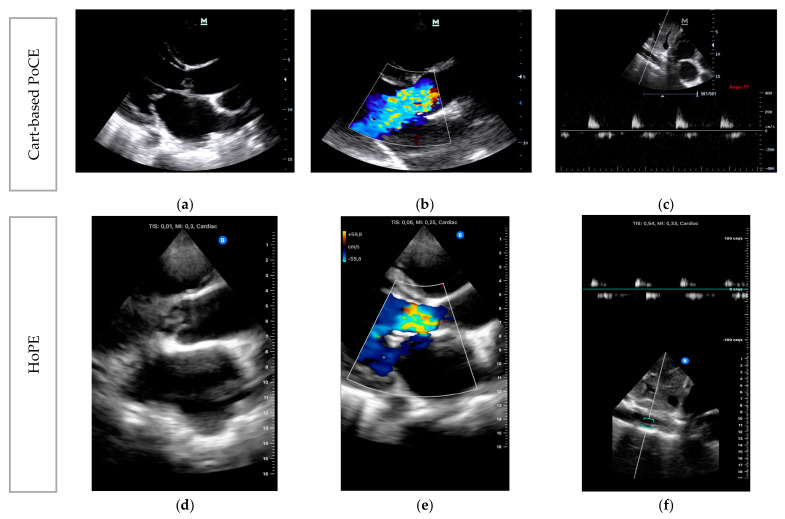
Case 2: Point-of-care echocardiography images at presentation: (**a**) 2D B-mode of the PLAX; (**b**) CFD of the PLAX; (**c**) PWD of the abdominal aorta. Cart-based images captured on Mindray Premium M7 (**a**–**c**) and handheld images captured on Butterfly iQ+ (**d**–**f**).

**Figure 7 diagnostics-13-02581-f007:**
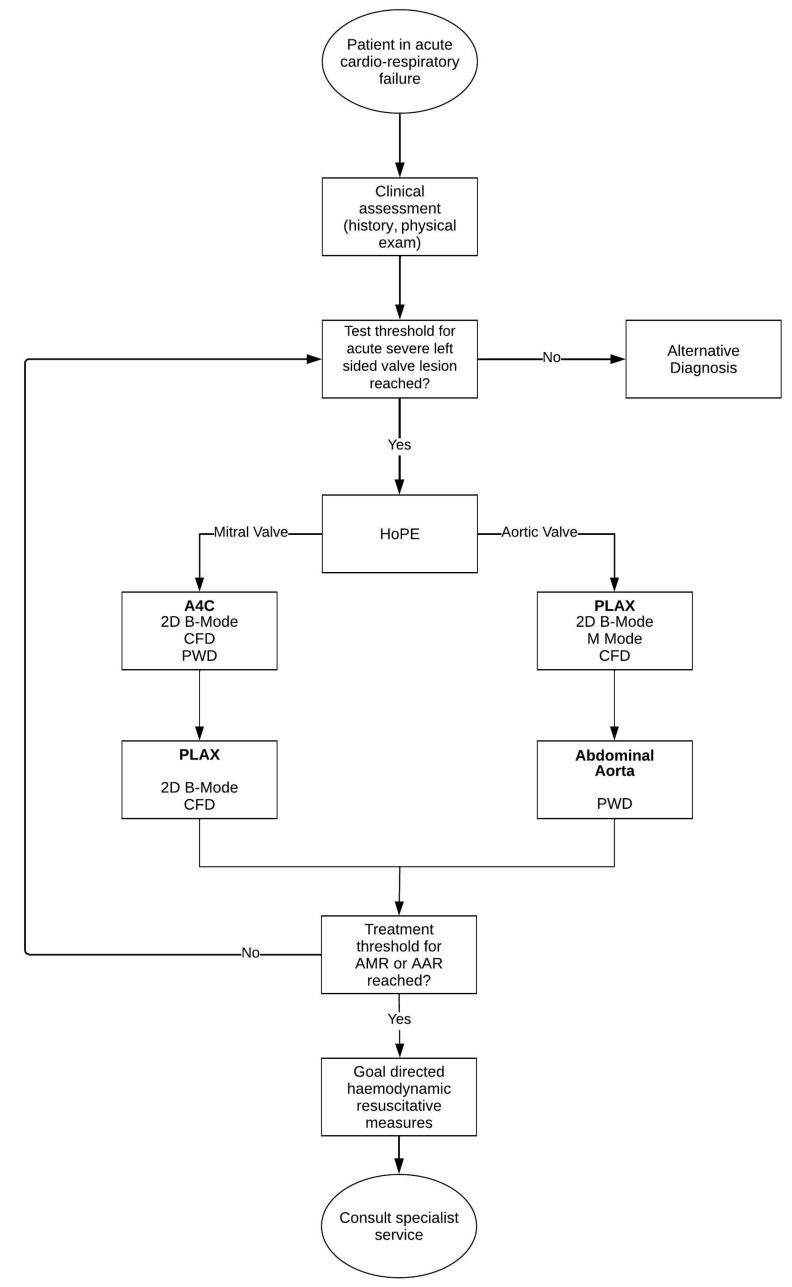
Novel diagnostic framework incorporating HoPE for the diagnosis of acute severe left-sided emergencies in patients presenting with acute undifferentiated cardiorespiratory failure.

## Data Availability

Not applicable.

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
