# Peer review of "Establishing a Novel Diagnostic Framework Using Handheld Point-of-Care Focused-Echocardiography (HoPE) for Acute Left-Sided Cardiac Valve Emergencies: A Bayesian Approach for Emergency Physicians in Resource-Limited Settings"

_diagnostics, 2023, doi:10.3390/diagnostics13152581_

Round 1

Reviewer 1 Report

Firstly I’d like to congratulate- very interesting approach and methods (Bayesian logic). Secondly, the review is well written and all known and important references in this field were appropriately cited. 

Author Response

Thank you for your insightful and encouraging review

Reviewer 2 Report

The article is a narrative review on left sided cardiac valve disease, namely acute aortic regurgitation and acute mitral regurgitation, including the importance of using handheld devices for performing point-of-care ultrasound (PoCE/PoCUS) in such cases. The objective is to suggest a diagnostic framework that could potentially be used as a protocol for early diagnosis of acute valve disease, utilizing Handheld Point-of-Care Echocardiography (HH-PoCE) as a diagnostic tool.

This topic is relevant for low-resource settings, as it is intended to provide guidance to clinicians, especially towards non-cardiologists. The existing literature already includes articles and reviews regarding the use of HH-PoCE, and even protocols intended for widespread use, which can be successfully applied to low-resource emergency departments. Already cited in the current manuscript is the following clear and in-depth review on this topic -  DOI: 10.1097/EC9.0000000000000066, and by analysing all current POCUS protocols there seems to be a gap in the literature regarding the acute aortic and mitral regurgitation diagnosis.

However, I have some comments/suggestions:

- Title: Could be improved to underline that the manuscript’s goal is the new diagnostic protocol, not the review in itself, and that this protocol targets mostly non-cardiologists, such as emergency department doctors

- Abstract: does not clearly state the fact that current emergency medicine protocols do not include a more in-depth analysis and quantification of the severity of an acute valvular dysfunction.

- Introduction:

1. The gap in the literature which this manuscript addresses is not clearly explained by the authors.

Some potential ideas: Current protocols intended for managing patients with haemodynamic instability and shock already include the recommendation to search for valvular failure, such as the GDE protocol (PMID: 23032454). GDE only includes a color Doppler analysis, which inherently has limitations when it comes to quantifying the severity of regurgitation. Moreover, the ACES protocol (PMID: 19164614) intended for approaching cases of undifferentiated hypotension does not include a PoCUS evaluation of heart valves. Existing protocols which include PoCUS evaluation are not updated to include a more in-depth and accurate quantification of the severity of valve dysfunctions. This is where the current manuscript may be positioned to offer some changes and improvements in existing emergency medicine protocols.

2. It may be of use to state that the framework suggested in this manuscript could be better used by integrating it in already existing protocols that are already routinely used and well-known by clinicians.

3. Clinical case #1 Previous myocardial infarction and already diagnosed mitral valve regurgitation should already provide an indication for ultrasound re-evaluation. It is also not very clear what the aetiology for the mitral valve regurgitation was, at the first presentation. The eccentric regurgitant jet already should raise some questions.

Maybe the article could be more clear in highlighting the importance of a more in-depth valve assessment using HH-PoCE devices by applying existing protocols (RUSH, ACES, GDE), then proving that the addition of the suggested new framework adds a proper evaluation of the valves and leads to a correct early diagnosis of the primary issue (acute valve failure), that would otherwise be missed or diagnosed too late. It is important to state that new HH devices already have spectral doppler capabilities and valvular assessment does not significantly prolong the duration of existing PoCE/POCUS protocols, or maybe that the benefits clearly outweigh the prolonged examination time.

4.  Line 337: “If this estimate breaches the test threshold - proceeds to evaluation with HH-PoCE.” The fact that test threshold for acute severe left sided valve lesion is not defined in the flowchart, may lead to confusions.

5. Lines 363-365 This premise, although relevant for some low-resource settings is likely to be invalidated by new and affordable handheld devices that are emerging on the market, that include spectral Doppler capabilities, even CW Doppler.

6. The strengths and limitations of the manuscript are not properly addressed. Usually, protocols and flowchart-type approaches are introduced only after enough evidence regarding their efficiency was provided based on trials or vast amounts of statistical data collected from registries. The authors should properly address this limitation.

7. The manuscript is not very clear in stating that the suggested framework cannot function as a standalone diagnostic protocol, because it addresses only left sided valve lesions.

8. Authors may improve the traction of such a framework by referring to existing articles which prove that even inexperienced users of ultrasound devices can be easily trained and that they may achieve good accuracies in detecting valve diseases after a few training sessions. This could be clearly preferred over clinicians who may regard their inexperience as a prohibitive factor in using such HH-PoCE devices.

- Conclusion: May benefit from some clarification and improvements on the value of the suggested diagnostic framework and where it may belong in the current setting. 

The English language needs minor adjustments.

Author Response

Thank you for your insightful and thorough review. Please see the attachment for our responses.

Reviewer 3 Report

Major feedback:

The authors present a possibly novel framework for using point-of-care cardiac ultrasound (cardiac POCUS) to screen for acute left-sided valvular insufficiency. They use two cases and a flow-chart to suggest how this framework could be used systematically by other clinicians.

I think the author’s suggestions are interesting and may be novel, but I think some additional work is required to make this useful to other clinicians.

1)    Figure 1 - please define precisely what you mean by “Test threshold for severe acute left-sided valvular dysfunction.”

2)    Figure 1 – please define what criteria you want cardiac POCUS users to use to “rule in” acute left-sided valvular insufficiency?  One option is to take the ASE guidelines defining cutoffs of various grades of severity and encourage cardiac POCUS users to considers any left-sided valve insufficiency that appears moderate or greater to be potentially severe (given the inherent limitations of POCUS devices being used for answering challenging questions like grading severity of valvular disease, I would think that a low threshold for ruling in significant disease should be used to maximize the sensitivity of the exam.  If you disagree, please explain why).

3)    Cases 1 and 2: If you’re suggesting that your approach is Bayesian, can you illustrate the use of Bayes theorem in your manuscript for each of your 2 cases?  E.g., you can suggest some arbitrary probabilities at each step (and acknowledge these probabilities to be arbitrary) to illustrate the point.

4)    It would help if you define a few terms more explicitly, such as “cardiac POCUS”.  Cardiac POCUS has varying definitions depending on who is using the term in medicine.  But for me, cardiac POCUS is best defined as broadly as a cardiac ultrasound exam (usually a transthoracic one) that is performed and interpreted by a patient’s primary treating provided.  This is contrast to a consultative cardiac ultrasound: which is requested by a patient’s primary treating provided but performed by a separate specialistic team (usually performed by a cardiac sonographer and interpreted by an echo-cardiologist).  Within cardiac POCUS, there are at least two subcategories: (1) focused cardiac ultrasound (FoCUS) and (2) point-of-care echocardiography (PMID: 35582953).  Your paper is clearly referring to point-of-care echocardiography rather than FoCUS.  Although you acknowledge the existence of FoCUS in your paper once (line 144), I would argue that you use this term in a way that is at odds with ASE guidelines.  Per ASE guidelines, FoCUS is NOT a form of point-of-care echocardiography because transthoracic echocardiography requires all of the following: (a) person performing the exam to have comprehensive knowledge of transthoracic cardiac ultrasound image acquisition; (b) person interpreting the exam to have comprehensive knowledge of transthoracic cardiac ultrasound image interpretation; (c) exam to include both greyscale ultrasound and advanced imaging modes (EKG gating, color/spectral Doppler); and (d) the exam should answer questions quantitatively.  In contrast a FoCUS exam: (a) can be performed by someone with training in a subset of cardiac ultrasound views; (b) interpreted by someone with training in only a subset of cardiac ultrasound pathologies; (c) requires only greyscale ultrasound; and (d) answers questions qualitatively.  So FoCUS is NOT a subset of “point-of-care echocardiography”, but rather both FoCUS and point-of-care echocardiography are subset of cardiac POCUS (the general term to describe a cardiac ultrasound exam performed and interpreted by a patient’s primary treating provider) (PMID: 35582953).  I would encourage the authors to use the term “echocardiography” judiciously in accordance with published ASE guidelines (e.g., PMID: 21111923; 23711341; 24951446)

Minor feedback:

Line 266 - Case 1: Your description of the case is confusing.  During the patient’s initial presentation, it appears that there was already sonographic data to support the diagnosis of acute severe MR based on the image you provide.  So I would argue that, during the patient’s re-presentation, the “pre-test probability of AMR” was not just “high” but already established (i.e., 100%).  Bayes Theorem doesn’t make sense as a concept when, in a 2-test sequence, the initial test provides the diagnosis.  One could argue that the problem here is that the diagnosis was present at the time of discharge from index hospitalization and was simply missed by the clinical team.

Line 276 – This sentence contains redundant use of “one month”:  “One month after she was discharged in a stable condition - following a short stay in 276our coronary care unit after declining to be transferred to the dedicated cardiac unit at our 277quaternary referral hospital - she re-presented to PSRH-ED one month later in cardi- 278orespiratory collapse.”

Line 289 – Figure 5 – your (a)/(b) and (c)/(d) labels are reversed from the images themselves.

Author Response

Thank you for your thorough and valuable review in helping us improve the manuscript significantly. Please see attachment.

Reviewer 4 Report

The idea of using handheld point-of-care echocardiography for the quick diagnosis of acute left heart valve regurgitation is absolutely right, however, the review does not bring new scientifically and clinically significant knowledge. Review is too long and contains well-known and documented medical knowledge about acute mitral and aortic regurgitation. The presented schemes are only a reminder of the pathophysiology of valvular defects, also the echocardiographic methodology is widely known and used.

Author Response

Thank you for your comments and concerns regarding our manuscript. We have revised the manuscript to reflect the context it should be viewed more accurately.

Limited resource settings often lack access to cardiologists (like our service). They may not be fortunate enough to have the finances to procure cart-based echocardiography capable machines - given the competing interests in the health goals of developing nations. Handheld ultrasound devices have emerged as lower-cost alternatives - potentially giving these patients access to POCUS-guided management through trained emergency physicians. Current POCUS applications and protocols include the heart and its FOCUS applications but fail to offer an assessment of acute left-sided valvular pathology - which is the devastating cause of acute cardiorespiratory failure. As illustrated in just two cases in the same year, the utility of an exam structured enough to improve diagnostic evaluation of the left-sided valves may improve the diagnosis and subsequent management of these conditions.

Leveraging Bayesian logic and medical diagnostics, we hypothesise that our framework can be used and/or integrated into other well known POCUS protocols to assist in the assessment of left sided valve disease in resource poor settings where advanced cardiac imaging services are not available.

The manuscript has also been updated to include a review of studies where POCUS was studied in patients presenting to the emergency department with hypotension - but due likely to the lack of a structured approach to the limited but sufficient quantification of left-sided valve pathology - they reported no cases of valvular abnormalities despite studying over 250 patients across north America and south Africa (PMID: 32968565).

Considering that our intended audience is non-cardiologists working in non-tertiary centres, we offered an in-depth review of pathophysiology - supporting our claim that physical examination is inadequate for AAR and AMR - before describing medical diagnostics and our diagnostic framework - given that this audience may be unfamiliar with it.

Nomenclature and terminology have been updated to better align with ASE and ESC guidance, and the title has been improved to better suit the core message of the paper. More limitations have been added, and a primer for further research into the real-world application.

We would still like to thank you for your review and trust that these modifications and explanations to your concerns will be satisfied.

Round 2

Reviewer 3 Report

The authors have addressed all of my feedback and I think the paper is now worthy of publication.

Reviewer 4 Report

I stand by my previous review. The manuscript is too long, and after the review it got even longer. It would be necessary to decide whether it is a review or a description of clinical cases. Instead of discussing the commonly known pathophysiological phenomena in valvular diseases, it would be better to include in the methodology the content contained in the response to the reviewer.